# Teachers' Workplace Victimization, Job Burnout, and Somatic and Posttraumatic Symptoms: A Structural Equation Modeling Examination

**Mahira Ghadban \*, Ruth Berkowitz and Guy Enosh**

School of Social Work, University of Haifa, Haifa 3498838, Israel
* Correspondence: mahira.ghadban@gmail.com; Tel.: +972-509070072

**Abstract:** Teachers' workplace victimization is a serious and concerning problem known to have numerous negative occupational and health outcomes for teachers. Surprisingly, however, the scientific literature has broadly overlooked schools as settings in which serious workplace violence occurs, and it has failed to systematically explore the antecedents and consequences of workplace victimization for teachers. To fill this gap, this study examined the structures of associations of teachers' victimization via threats, verbal violence, and property violence with somatic and posttraumatic symptoms using structural equation modeling. The sample included 366 Arabic and Jewish teachers in Israel. The findings indicate that the teachers who reported victimization and symptoms experienced burnout, whereas the teachers who reported victimization and had no symptoms did not experience burnout. A significant relationship of teachers' personal and professional characteristics with victimization, symptoms, and burnout also emerged. These findings advance our theoretical understanding of the predictors and consequences of violence against teachers in schools. Useful practices to improve teachers' occupational and health outcomes are discussed.

**Keywords:** workplace victimization; somatic symptoms; posttraumatic symptoms; burnout

## 1. Introduction

Teachers' workplace victimization (TWV) is a serious and concerning problem prevalent in various countries throughout the world [1–4]. The literature has identified individual and organizational factors that may contribute to TWV, including workload; conflict; and a lack of support among school staff members, seniority, and teacher gender [5–13]. TWV is known to have many negative consequences for teachers in terms of occupational (e.g., burnout) and health outcomes (e.g., somatic and posttraumatic symptoms). In addition, TWV impairs the healthy development and learning of students [9,14–20]. Although extreme events of TWV gain significant attention in the media and increased public concern, less serious events occur more frequently, even daily, and impair teachers' functioning, emphasizing the importance of addressing this issue [21,22]. Given this background, it is important to examine the associations between teachers' victimization and somatic and posttraumatic symptoms and burnout. This study employed structural equation modeling (SEM) to examine the structure of associations between TWV and antecedents of TWV; teachers' victimization via threats, verbal violence, and property violence; and somatic and posttraumatic symptoms. The findings can advance our theoretical understanding of the predictors and consequences of violence against teachers in schools in order to inform useful practices to improve teachers' occupational and health outcomes and students' healthy development [23].

### 1.1. School Violence and Teachers' Workplace Victimization

School violence is a serious and persistent problem. All members of a school community can be victims of violent acts, including teachers [1,2,4]. TWV may include physical

(e.g., pushes and kicks) and verbal (e.g., curses, humiliation, and threats) violence, stealing and the damaging of property, and electronic-media-related harm (cyberbullying) [4,21,24–26]. These types of school violence behaviors may be organized by severity to discriminate between mild and moderately aggressive behaviors and more serious forms of violence. Milder forms of violence are more prevalent and may include verbal abuse, threats, and property aggression, whereas more severe violence is rare and may include physical and sexual violence [20,27,28]. Because less serious forms of violence are more prevalent and experienced by many teachers on a daily basis [21,22], this study focused on the milder forms of teachers' victimization: verbal abuse, threats, and property violence.

### 1.2. Negative Occupational and Emotional Outcomes of TWV

Studies have indicated that workplace victimization and TWV are associated with negative occupational and emotional outcomes. Research has indicated negative occupational outcomes among teachers who have experienced workplace victimization, including greater job burnout [29–31]—a psychological syndrome of emotional exhaustion at work and cynicism toward work, which occurs frequently among individuals who work with other people in some capacity [32]. Job burnout can increase negative physiological (fatigue, headaches, and hypertension), emotional (emotional exhaustion, depression, and anxiety), behavioral (insomnia, a decline in performance, social withdrawal, and other interpersonal difficulties), and cognitive (self-doubt, guilt, and a sense of disillusionment) symptoms [32,33].

Furthermore, teachers' victimization is associated with posttraumatic symptoms [19,20,31]. Posttraumatic stress disorder (PTSD) is a mental disorder that occurs when a person is exposed to a traumatic event that involves danger to them or others [34]. A meta-analytic review has demonstrated that PTSD is associated with somatic symptoms, including greater frequencies and severities of musculoskeletal, cardiorespiratory, gastrointestinal, and generalized physical complaints [35]. Indeed, workplace violence studies have revealed an association between workplace victimization and somatic symptoms among workers [14,20,36].

### 1.3. Associations of Teachers' Workplace Victimization, Posttraumatic Symptoms, and Burnout

The scientific literature linking TWV, posttraumatic and somatic symptoms, and burnout has revealed inconclusive evidence. Prior studies have indicated a direct association between TWV and burnout [37–44]. Teachers experiencing workplace victimization have also been found to be more likely to be absent from school and leave the teaching profession—clear indications of job burnout [1,20,25,41,42].

Other studies have suggested that teachers' and other professionals' workplace victimization is associated with somatic symptoms, PTSD symptoms, and burnout [9,14–20,43,44]. Thus, we hypothesized that TWV is directly associated with symptoms and burnout.

Other studies have suggested an indirect association between workplace victimization and burnout via the mediation of posttraumatic and somatic symptoms (e.g., Yang [38]). To illustrate, Berg and Cornell [5] identified an association between teacher-directed violence and burnout, with some burnout explained by distress symptoms. Similarly, Rojas-Flores and colleagues [31] indicated a significant indirect effect of teachers' exposure to violence on burnout through PTSD symptomatology across two independent samples of teachers in Central America. However, a different study indicated a nonsignificant association between burnout and symptoms, suggesting that teachers' victimization via occasional and severe bullying predicted psychological distress, although burnout did not mediate those associations [39]. These inconclusive and contradictory findings demand additional investigation. Thus, we further tested the indirect association between TWV and burnout via the mediation of symptoms.

### 1.4. Teachers' Background Characteristics

Prior research has noted numerous factors that may contribute to TWV, including teachers' individual characteristics, the degree of conflict among colleagues, workload, and the ethnocultural background of the school community.

### 1.4.1. Personal Characteristics

Some studies have attempted to identify the relationship between teachers' personal characteristics, such as age, gender, and seniority, and exposure to violence. The findings are contradicting and inconsistent [5–7,9,12,40,45]. Some researchers have further indicated disparities in victimization rates among female and male teachers across types of school violence [4,21,26,46]. Inconsistent results have been found regarding teachers' level of education and seniority, tenure in their school, and chances of workplace victimization [6,7,12]. Additional research is needed to explore the associations between teachers' personal characteristics and their likelihood of victimization.

Prior research has also examined the contribution of teachers' racial or ethnic origin to workplace victimization. Research in the United States has shown inconsistent findings [12,21,45,46]. In Israel, where Arabic- and Hebrew-speaking teachers work in separate school systems, researchers have identified distinct patterns of TWV via specific types of violence across Hebrew- and Arabic-language schools. Whereas TWV via verbal violence has been found to be more prevalent in Hebrew-language schools, TWV via threats, property damage, or theft has been found to be higher for teachers in Arabic-language schools [47]. Different victimization patterns across cultural groups may be attributed to sociopolitical aspects associated with culture, norms and heritage, belief systems, poverty, deprivation, power imbalances, political oppression, and a lack of opportunities for education and employment [48–51].

### 1.4.2. Professional Characteristics

Teacher's professional characteristics, such as collegial conflict and workload, have also been found to be associated with workplace victimization [1,45]. Prior research has found a positive association between negative interpersonal relationships among employees and workplace victimization [52,53]. In studies in different work settings, associations have been found between positive interpersonal relationships at work and innovative behaviors and being more involved in work [54,55]. Bullying by a supervisor or coworker can have serious negative consequences, including a sense of loneliness, isolation, and greater impotence, and it can interfere with a sense of commitment to work activities [53]. Prior research has further revealed that conflictual relationships between teachers and principals can increase work absenteeism, workload, and dissatisfaction and intolerance among teachers [11].

Prior research has also examined the contribution of teachers' workload and the chances of workplace victimization. Job workload is measured by demands from management or those in charge [54]. An increased job workload has negative consequences for teachers, including greater emotional exhaustion [10,13,19] and decreased job satisfaction, commitment, performance, and motivation [8,10]. Somatic and posttraumatic symptoms may also stem from conflict between staff members and workload [8,10,11,13,31].

### 1.4.3. The Effect of the COVID-19 Outbreak on Schools

Since December 2019, nothing has been more important on the world's sociopolitical and economic agenda than the COVID-19 pandemic [56]. Reducing physical contact has been the most common strategy adopted by governments to reduce the spread of COVID-19. This led most countries around the world to close their schools for periods of time [57–61]. Research among students conducted immediately after the COVID-19 global breakout has revealed a high percentage of depression and anxiety [62–64], posttraumatic symptoms [65,66], sleeping disorders [66], psychological distress [65,67], and loneliness [68] and an overall drop in mood and life satisfaction [69] among youth.

Nevertheless, students were not the only people who demonstrated negative outcomes following the pandemic outbreak. Some limited research has indicated a decreased functioning and well-being of teachers following the outbreak [70]. Moving to remote teaching, social distancing, having to adopt new teaching skills and methods, and having to deal with increased behavioral and emotional problems of students promoted high levels

of stress [71–73] and burnout [74–77] and decreased intention to stay among teachers [63]. This study was conducted in spring 2020 during the first and second lockdowns in Israel, in which schools closed and reopened intermittently.

### 1.5. Israeli Education System

The Israeli education system reflects the cultural and ethnic diversity of Israeli society. Prior to the establishment of Israel in 1948, various political parties established schools based on their distinct political domains, separating students by ideologies related to religion, culture, socialism, and language [78]. Such an approach, which currently prevails, emphasizes maintaining a distinct language, curriculum, and instruction paradigm for each subpopulation and creating a culture and mission that match the students and their families, with a focus on culture and religion. Therefore, language and culture are defining characteristics for public schools in Israel, such that students from Arab Muslim, Bedouin, Christian, Druze, and Jewish families, whether secular or religious, attend schools that coincide with their ethnocultural and language preferences.

In Israel, most teachers are of the same ethnic origin as their students, such that Arabic-speaking teachers work in Arabic-language schools and teachers from Jewish society teach in Hebrew-language schools. Thus, in the current research, the ethnocultural origin of teachers coincided with the ethnocultural origin of the students and school community.

In summary, prior findings have demonstrated inconclusive evidence regarding the associations among teachers' background characteristics, conflictual work relationships, TWV, somatic and posttraumatic symptoms, and job burnout. Because the structures of the relationships between the variables remain unclear, this study sought to fill this gap by estimating and testing the structures of pathways among TWV and related outcomes, as presented in a theoretical model that was developed based on the existing literature. Figure 1 presents the full study theoretical model.

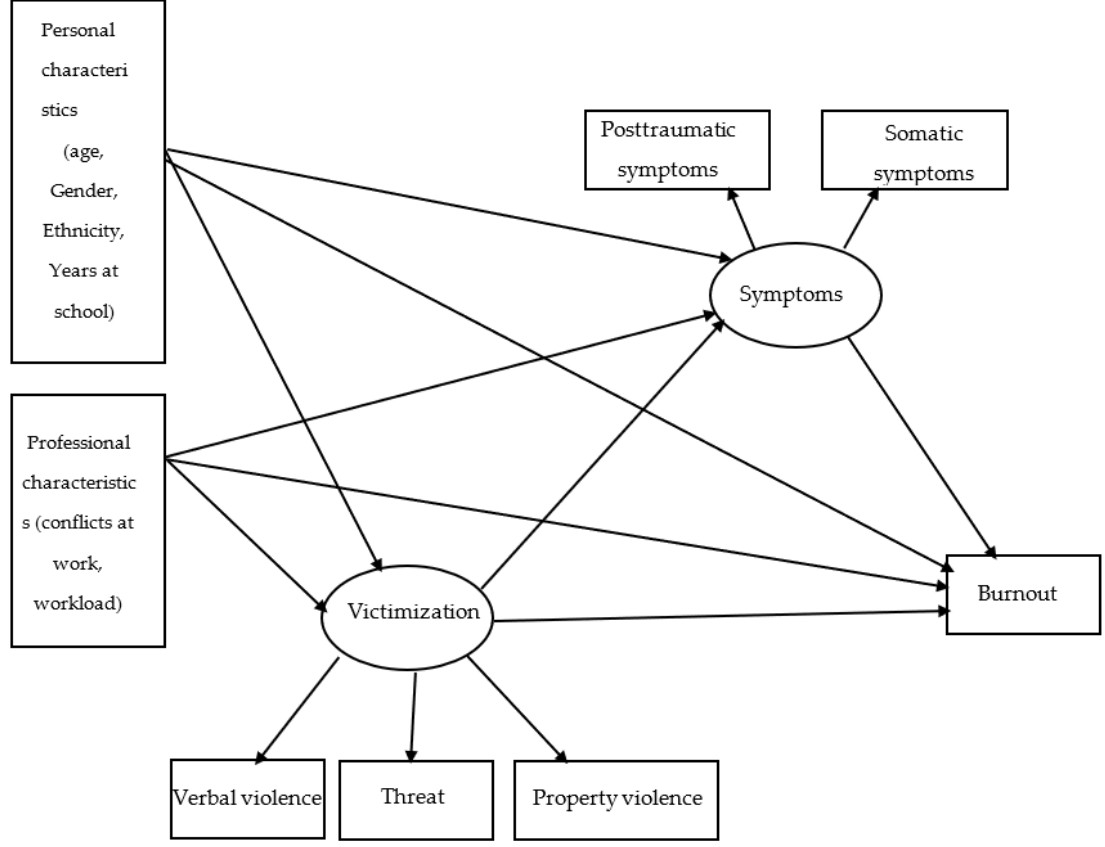

**Figure 1.** Full study model.

A theoretical model visually displays structural relationships, typically based on equations connecting conceptual variables, to formalize a theory. SEM involves statistical methods that estimate relationships among observed variables that represent conceptual variables in statistical models [79].

The analyzed model included eight variables. Teachers' personal characteristics included ethnicity, age, gender, and years at school. Teachers' professional characteristics included workload and conflict at work; victimization via verbal abuse, property damage, and threats of violence; and somatic and posttraumatic symptoms. The model indicated an association between teachers' background variables and professional characteristics and victimization. Associations were also present between burnout, victimization, and symptoms. All eight variables were associated with burnout.

The following hypotheses were tested while controlling for teachers' background characteristics:

1. An association exists between teachers' victimization and burnout, such that teachers who experience more victimization exhibit greater burnout.
2. Somatic and posttraumatic symptoms mediate the association between teachers' victimization and burnout.

## 2. Materials and Methods

Data for the current research were collected between December 2019 and May 2020, a period in which the COVID-19 pandemic broke out in Israel and schools closed and re-opened intermittently. We used snowball convenience sampling via social media platforms to locate and recruit participants for the study. The researchers shared the questionnaire via social media and asked participants to pass it on to their teacher acquaintances. The sample included 363 teachers from both Hebrew and Arabic schools in Israel. This study was approved by the ethics committee of the authors' affiliated institution.

### 2.1. Measurements

2.1.1. Teachers' Burnout ($\alpha = 0.792$)

Teachers' burnout was measured using the Maslach Burnout Inventory Educators Survey, a teacher-specific version of the inventory [32]. The scale consists of 22 items measuring the frequency of experiences related to burnout (e.g., "I feel emotionally drained from my job"; "I feel exhausted at the end of my work day"). The responses were given on a 7-point Likert scale (0 = *never* to 6 = *every day*). Job burnout was computed as the average score of these 22 items (*M* = 4.08). Higher scores indicated greater job burnout.

2.1.2. Teachers' Workplace Victimization

TWV was measured using a modified version of the Client Violence Questionnaire [80]. For the present study, we used 10 questions that measure verbal violence (e.g., "a student and/or parent shouted at you"; "a student and/or parent has cursed you"; $\alpha = 0.910$); property violence (e.g., "a student and/or parent threw an object on the wall or floor"; "a student and/or parent dropped objects, furniture, or kicked furniture in the classroom or in the counselor's room"; $\alpha = 0.838$); and threats (e.g., "a student and/or parent threatened to damage your property"; "a student and/or parent used a general threat (for example: 'You will be sorry ...')"; $\alpha = 0.746$). The teachers were asked to indicate the frequency with which they experienced violence in the past 3 months. The responses were given on a 7-point Likert-scale (0 = *never* to 6 = *six times or more*). Teachers' victimization was computed as the average of these 10 items.

2.1.3. Posttraumatic Symptoms ($\alpha = 0.969$)

Posttraumatic symptoms were measured using the Posttraumatic Symptoms Questionnaire [81]. This questionnaire includes 17 items that measure the prevalence of experiencing various symptoms typical of trauma as a result of aggression during the past month (e.g., "Did you re-experience the aggressive event and act or feel like it was happening again?"; "Do you try to avoid activities, people, or places that remind you of the aggressive event?"). The

responses were given on a 5-point Likert scale (1 = *never* to 5 = *daily*). The mean of these 17 items was used for subsequent analyses.

### 2.1.4. Somatic Symptoms ($\alpha = 0.939$)

Somatic symptoms were measured using the Somatic Symptoms Questionnaire [48]. In the present study, we used nine statements to measure the somatic symptoms experienced during the past month (e.g., "Did you experience an accelerated heartbeat?"; "Did you experience abdominal pain?"). The responses were given on a 5-point Likert scale (1 = *never* to 5 = *daily*). A composite measure was computed by averaging the responses.

### 2.1.5. Collegial Conflicts ($\alpha = 0.788$)

Collegial conflicts were measured using the Interpersonal Conflict at Work Scale [54]. This scale consists of four items that measure the frequency of interpersonal conflicts at work (e.g., "How often do you get into arguments with others at work?"). The responses were given on a 5-point Likert scale (1 = *less than once per month or never* to 5 = *several times per day*). A composite measure was computed by averaging the responses.

### 2.1.6. Workload ($\alpha = 0.845$)

Workload was measured using the Quantitative Workload Inventory [54]. The participants were asked to indicate their level of agreement with five items describing workload experiences (e.g., "I have to deal with increased demands from management and the Education Ministry"; "I do not have enough time to do all that is required in my position"). The responses were given on a 7-point Likert scale (1 = *not at all* to 7 = *very much*). A composite measure was computed by averaging the responses.

### 2.1.7. Sociodemographics

The participants provided general demographic information on their age, gender, education level (high school education, BA, MA, or higher degree), ethnicity (Jewish or Arabic), and number of years working at their current school.

### 2.2. Analytic Method

The relationships between variables were examined by carrying out correlation tests. We employed SEM to test the hypothetical model of the causal structure among the study variables based on the existing literature. This technique allowed for the consideration of a few dependent variables, thus making the latent variable structures more reliable than using observed variables and including measurement errors.

This study employed SEM to examine the structure of the relationships between TWV and symptoms and burnout. Because the model was based on correlations, it could not demonstrate causality and could only indicate if the approximate causal model matched the patterns of relationships in the data. The data were analyzed using IBM SPSS software and SAS version 9.4 [82].

## 3. Results

### 3.1. Sample Characteristics

The sample included 366 teachers; most identified as female (81.8%). Half of the participating teachers identified as Arab (56.4%), less than half identified as Jewish (43%), and a small proportion (0.6%) did not indicate their ethnicity. Most participants (57%) had a master's degree or higher. Less than half of the respondents (41.9%) reported having a bachelor's degree. Table 1 presents the means (and standard deviations) and correlations among the study variables.

**Table 1.** Means, standard deviations, and Pearson correlations among the study variables (*N* = 363).

| Variable | *M* | *SD* | 1 | 2 | 3 | 4 | 5 | 6 | 7 | 8 |
|---|---|---|---|---|---|---|---|---|---|---|
| 1. Burnout | 4.08 | 0.80 | | | | | | | | |
| 2. Verbal violence | 1.61 | 1.90 | 0.367 ** | | | | | | | |
| 3. Property violence | 1.45 | 1.77 | 0.331 ** | 0.719 ** | | | | | | |
| 4. Threats | 0.57 | 0.98 | 0.231 ** | 0.626 ** | 0.597 ** | | | | | |
| 5. PTSD symptoms | 1.53 | 0.90 | 0.428 ** | 0.446 ** | 0.377 ** | 0.340 ** | | | | |
| 6. Somatic symptoms | 1.61 | 0.97 | 0.471 ** | 0.451 ** | 0.414 ** | 0.367 ** | 0.909 ** | | | |
| 7. Work conflict | 1.63 | 0.75 | 0.327 ** | 0.316 ** | 0.250 ** | 0.252 ** | 0.429 ** | 0.428 ** | | |
| 8. Workload | 5.34 | 1.35 | 0.575 ** | 0.452 ** | 0.384 ** | 0.240 ** | 0.292 ** | 0.328 ** | 0.249 ** | |
| 9. Tenure at school | 10.44 | 8.88 | −0.079 | −0.188 ** | −0.137 * | −0.043 | −0.186 ** | −0.169 ** | −0.124 * | −0.082 |

\* $p < 0.05$. \*\* $p < 0.01$.

### 3.2. Associations among Study Variables

Figure 2 presents the study model results. To simplify and clarify the relevant effects for the research hypothesis, the model does not present the effects of the background variables. The model's fit indexes were good (GFI = 0.9817, RMSEA = 0.0379, CFI = 0.9929, and TLI = 0.9813).

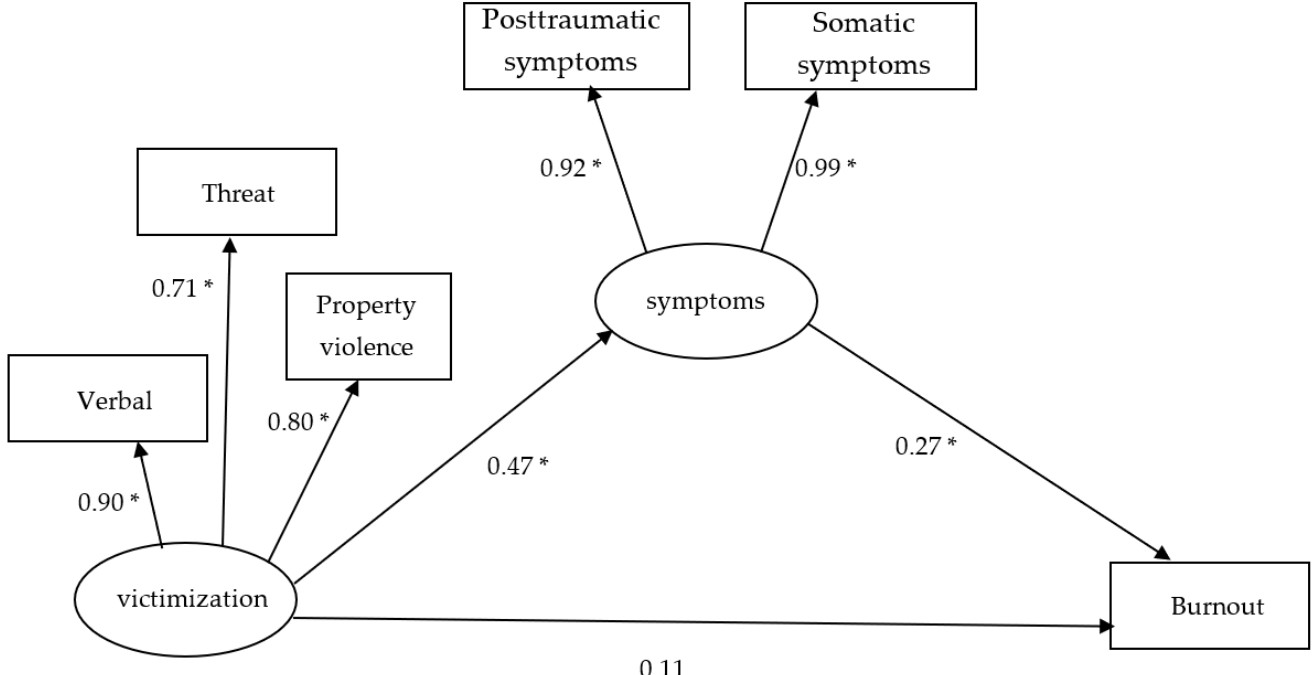

**Figure 2.** Standardized coefficients of the model and significance tests. \* $p < 0.001$.

We found that the teachers who experienced victimization and had symptoms also experienced burnout (β = 0.27, $p < 0.001$; see Table 2). However, teachers who experienced victimization without symptoms did not experience burnout (β = 0.11, $p = 0.10$). That is, we found support for one of our two main hypotheses.

**Table 2.** Standardized and unstandardized path results.

| Path | β | *b* | SE | *t* | *p* |
|---|---|---|---|---|---|
| Theoretical variables | | | | | |
| Victimization → burnout | 0.105 | 0.062 | 0.038 | 1.630 | 0.103 |
| Victimization → symptoms | 0.470 | 0.270 | 0.036 | 7.459 | <0.001 |
| Symptoms → burnout | 0.265 | 0.273 | 0.055 | 4.937 | <0.001 |

**Table 2.** *Cont.*

| Path | β | *b* | SE | *t* | *p* |
|---|---|---|---|---|---|
| Control variables | | | | | |
| Age → victimization | 0.048 | 0.006 | 0.009 | 0.743 | 0.457 |
| Age → symptoms | 0.050 | 0.004 | 0.004 | 0.821 | 0.411 |
| Age → burnout | −0.085 | −0.007 | 0.004 | −1.556 | 0.119 |
| Gender → victimization | 0.039 | 0.139 | 0.181 | 0.768 | 0.442 |
| Gender → symptoms | 0.013 | 0.025 | 0.098 | 0.262 | 0.793 |
| Gender → burnout | 0.002 | 0.005 | 0.090 | 0.062 | 0.950 |
| Ethnicity → victimization | −0.264 | −0.715 | 0.143 | −4.990 | <0.001 |
| Ethnicity → symptoms | 0.160 | 0.249 | 0.083 | 3.005 | <0.001 |
| Ethnicity → burnout | 0.171 | 0.275 | 0.079 | 3.455 | <0.001 |
| Years at school → victimization | −0.144 | −0.021 | 0.009 | −2.241 | 0.025 |
| Years at school → symptoms | −0.103 | −0.008 | 0.005 | −1.716 | 0.086 |
| Years at school → burnout | 0.073 | 0.006 | 0.004 | 1.350 | 0.176 |
| Conflicts → victimization | 0.221 | 0.438 | 0.100 | 4.368 | <0.001 |
| Conflicts → symptoms | 0.220 | 0.251 | 0.056 | 4.455 | <0.001 |
| Conflicts → burnout | 0.038 | 0.044 | 0.054 | 0.821 | 0.411 |
| Workload → victimization | 0.321 | 0.320 | 0.055 | 5.780 | <0.001 |
| Workload → symptoms | 0.100 | 0.058 | 0.031 | 1.848 | 0.064 |
| Workload → burnout | 0.479 | 0.283 | 0.029 | 9.825 | <0.001 |

### *3.3. Background Variables and Their Effects*

The teachers' ethnicities significantly affected victimization, such that Jewish teachers reported greater victimization than Arabic teachers ($\beta = -0.26$, $p < 0.001$). The teachers' ages ($\beta = 0.05$, $p = 0.46$) and genders ($\beta = 0.04$, $p = 0.44$) did not have significant effects on victimization. Years at school negatively affected victimization, such that the teachers with fewer years at school reported greater victimization than the teachers with more years at school ($\beta = -0.14$, $p < 0.05$). Workload and collegial conflict were significantly associated with victimization, such that the teachers who reported a greater workload ($\beta = 0.32$, $p < 0.001$) and conflict ($\beta = 0.22$, $p < 0.001$) experienced greater victimization.

In addition, the teachers' ethnicities ($\beta = 0.16$, $p < 0.001$) and conflict at work ($\beta = 0.22$, $p < 0.001$) significantly predicted symptoms. The findings show that the teachers from Arabic society reported more symptoms than the teachers from Jewish society. The findings also show that conflicts at work predicted symptoms, such that the teachers who reported conflicts at work had more somatic and posttraumatic symptoms. Ethnicity further predicted burnout ($\beta = 0.17$, $p < 0.001$), such that Arabic teachers reported greater burnout than Jewish teachers. In addition, workload predicted burnout ($\beta = 0.48$, $p < 0.001$), such that the teachers who reported a greater workload experienced more burnout.

### 4. Discussion

Despite extensive research on employees' sense of safety at work and workplace victimization in numerous workplaces in the private and public sectors, the scientific literature has broadly overlooked schools as settings in which serious workplace violence occurs. In order to fill this gap, this study utilized SEM to examine the structure of associations between TWV via threats, verbal violence, and property violence; somatic and posttraumatic symptoms; and burnout. The research model was based on prior research that examined the associations among workplace victimization, posttraumatic and somatic symptoms, and burnout [5,14,19,20,29–31,44].

Our main hypothesis was that teachers who experienced victimization would experience more posttraumatic and somatic symptoms and greater burnout. In contrast to prior scientific evidence indicating that workplace victimization predicts burnout [29–31], the current findings did not indicate these associations. A possible explanation for this may be that the negative consequences of victimization may lead victims to experience dissatisfaction with work and their workplace, prompting them to change schools or

leave the teaching profession; thus, reports on job burnout were not associated with victimization [1,20,25,26,29,41].

Our second hypothesis was that teachers who experienced greater victimization would report more posttraumatic and somatic symptoms. The current findings are consistent with prior research and meta-analytic reviews indicating that teachers and other employees reported increased symptoms and an increased risk of depression following exposure to violence [5,14,20,29,31,36,43,44,83,84]. Other studies have indicated a correlation between teacher-directed violence and burnout, with some of the burnout due to posttraumatic symptoms [5,31].

*4.1. Effects of Teachers' Background Characteristics*

The findings indicate that Jewish teachers experienced greater workplace victimization than Arabic teachers. These surprising finding are inconsistent with our hypothesis based on previous studies indicating a greater victimization of teachers in urban schools serving a high percentage of ethnocultural minority students than in schools catering to mainly nonminority populations [5,21,85]. These findings are especially surprising because they contradict prior research conducted in Israel, in which students reported that violence directed toward teachers occurred twice as much in Arabic-language schools than in Hebrew-language schools at all grade levels [3]. A possible explanation for this is that Arabic teachers are less likely to report being victimized by their students because Arabic society is collectivist and places great importance on gender, age, and status hierarchy [86,87] in contrast to Jewish society, which is described as similar to industrialized European and individualist egalitarian cultures, where gender roles and age hierarchy are less pronounced [87,88]. Hence, Arabic teachers are expected to have hierarchical authority by virtue of being a teacher, both in terms of age and their social role. These characteristics of Arabic society may make it more difficult for Arabic teachers to report violence perpetrated against them by their students. In accordance, the social desirability bias response suggests that, in research, socially desirable attitudes and behaviors are reported to avoid reporting socially undesirable attitudes and behaviors [89,90]. Indeed, previous studies have indicated that the pattern of social desirability bias is stronger and consistent in collectivist cultures, such as Arabic society, as compared to an individualist cultures, such as Jewish society [91,92].

A different explanation for the current findings is related to the fact that collectivist cultures tend to prefer strategies of compromise and problem solving, whereas individualist cultures tend to prefer strategies of goal-directed coercion. Compromise and problem-solving strategies are more likely to preserve relationships than goal-directed coercion strategies [93], and, as a result, they encourage less violence between teachers and students. Hence, it can be assumed that Arabic teachers maintain an affinity with students through compromising strategies, which lead to the teachers experiencing less victimization. This explanation is consistent with the findings of Martinez and colleagues [45], indicating that Latin American teachers in a collectivist culture experienced less violence than White teachers from an individualist culture.

In accordance with prior research [2,94], age and gender were not significantly related to TWV, although some previous studies have indicated that younger and female teachers experience greater workplace victimization [4,5,7,9,12,95,96]. In the current study, female teachers accounted for much of the sample (88%). The study sample characteristics coincide with the gender breakdown of the Israeli education system, in which female teachers constitute approximately 80% of all teachers [97]. Thus, the examination of differences in the type of violence against female teachers and male teachers is complicated.

In accordance with some prior research [94,96,98], the current findings suggest that the number of years working at the current school was negatively associated with TWV. It may be that teachers serving more years at their school are older than the teachers with less years at their schools. An alternative explanation is that the teachers working more years at school have better class management skills and, thus, experience less violence. The

classroom is an environment for potential violence because the interaction between teacher and student is more active, and the effectiveness of a teacher's classroom management skills is a strong indication of student violence toward teachers [24]. Teachers with more years of experience tend to be more capable of managing their class and student conflict [99], have a greater ability to deal with potential conflicts with students [100], and may have more control over student behavior [45]. As previously mentioned, teachers with greater seniority in school are usually older and generally enjoy greater respect from students because they project more confidence and are perceived as less vulnerable targets than younger and newer teachers [5,101]. In contrast to the current findings, other research has indicated a nonsignificant association between years at school and victimization [5–7,9,12]. Additional research could further explore the association between the number of years teachers have worked at their school, TWV, and other variables that could shed more light on this association, such as teachers' class management skills.

The finding that the teachers' ethnicities were significantly related to symptoms and burnout is interesting, as Arabic teachers reported more symptoms and burnout than Jewish teachers, although Jewish teachers reported more TWV. A possible explanation for these counterintuitive findings is related to the high levels of community violence in Arabic society in Israel. Prior research has indicated a greater exposure to community violence in Arabic society than in Jewish society, which has been found to lead to greater posttraumatic symptoms following exposure to violence among Arabic individuals [102]. Thus, it may be that Arabic teachers experience more symptoms than Jewish teachers not as a result of workplace victimization but rather due to a greater exposure to community violence. Additional research could further explore this hypothesis.

The findings of our study regarding job strain are consistent with those of previous studies that found that job strain predicted victimization and burnout [10,13,19,54]. Work conflict was significantly and positively associated with TWV and symptoms. That is, the teachers who were more involved in conflict with other teachers were also more involved in incidents perpetrated by parents and students, and they were more likely to suffer from posttraumatic and somatic symptoms. These findings are in accordance with previous studies that indicated an association between interpersonal conflicts and an increased likelihood of teachers' verbal victimization [52]. It is likely that short-term harassment can severely affect the victim, destroy their confidence, and interfere with their sense of commitment to work activities [53]. That is, interpersonal conflict and exposure to verbal violence can cause short-term damage, which may also psychologically harm the teachers. They are associated with structural issues and the managerial problems of creating a more beneficial atmosphere and climate among teachers and between teachers and parents and students.

To summarize the main conclusions of this study, a close association emerged between exposure to violence among teachers and exposure to violence from parents and students toward teachers. Until now, researchers had assumed that exposure to violence caused burnout. Our findings indicate that burnout is not a direct product of exposure to violence of one kind or another but rather a process mediated through symptoms.

### 4.2. Study Limitations and Recommendations for Future Research

Although informative, some findings and interpretations should be considered with the study limitations in mind. First, we used a snowball sampling method; thus, the study sample does not represent the entire population. Additional research is encouraged to further explore the associations among TWV, job burnout, and symptoms using larger and representative samples in order to generalize findings to the entire population of teachers and design effective prevention and intervention strategies. Second, the data for the study were collected at one point in time. These data demonstrate associations between variables rather than their causal relationships. To establish causal arguments based on large population-based samples, longitudinal data are required. Future studies

could employ longitudinal research arrays to explore causal arguments stemming from the relevant models and theories.

We further recommend examining more deeply ethnocultural disparities in TWV, antecedents, and outcomes across Arabic and Jewish societies and different roles, such as educators, managers, and professional teachers, to determine if differential relationships exist between roles in the two societies and burnout and symptoms. Additional research is encouraged to examine the associations among TWV, burnout, and symptoms across different cultures in order to uncover whether their associations manifest differently depending on the cultural context. Because school violence manifests differently across grade levels, additional research could explore the differences in the study model among teachers working in schools of all grade levels.

The current findings are unique in that they suggest that teachers who experience victimization and develop symptoms experience burnout more than teachers who do not develop symptoms. Nonetheless, because the data were collected during the outbreak of COVID-19 in Israel, educators likely experienced increased stress that affected the measurement. Thus, we encourage researchers to study TWV, job burnout, and symptoms in regular and less stressful periods. Further, because research has pointed to the critical roles of teachers' self-efficacy and self-esteem in times of crisis [103], we recommend including these variables in future studies.

*4.3. Practice Recommendations*

This study is the first in Israel to examine the issue of violence against teachers and the perceived consequences. This study examined the direct and indirect associations of TWV, suggesting that the experience of victimization predicted the appearance of symptoms that predicted burnout. This study adds to our understanding of teachers' personal perspective regarding the exposure of service providers, especially teachers, to the aggression of clients (students and parents), and the findings can inform intervention programs for schools that suit teachers' needs and that minimize the phenomenon of violence against teachers. Regarding implications at the policy level, violence needs to be reduced in both directions—that is, administrative intervention is needed to reduce violence, whether between staff members or between teachers and parents or students, in order to reduce the symptoms and suffering among teachers. Teachers should also be supported mentally through support groups, counseling and trauma treatment, mental first aid, and ongoing help.

**Author Contributions:** Methodology, M.G. and G.E.; formal analysis, M.G. and G.E.; investigation, M.G.; Resources, R.B.; writing—original draft, M.G.; writing—review & editing, R.B.; supervision, R.B. and G.E. All authors have read and agreed to the published version of the manuscript.

**Funding:** This research received no external funding.

**Institutional Review Board Statement:** The study meets the conditions of the committee for Ethical Research with Human Beings of the Faculty of Welfare and Health at the University of Haifa. Approval Code: 19/147.

**Informed Consent Statement:** Informed consent was obtained from all subjects involved in the study.

**Data Availability Statement:** Data are available through the first author.

**Conflicts of Interest:** The authors declare no conflict of interest.

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
