# Peer review of "Teachers’ Workplace Victimization, Job Burnout, and Somatic and Posttraumatic Symptoms: A Structural Equation Modeling Examination"

_education, doi:10.3390/educsci13020179_

Round 1
Reviewer 1 Report
Thank you for your effort to share and publish this article. I also found that this is your original works (Turnitin is kindly referred). However, there are several comments that need further clarification and improvements.
1. You intend to establish SEM but I did not find any prominent theory in your article to support your conceptual/theoretical framework (Kindly refer Hair et al. & etc.)
2. Your 2nd hypothesis- ‘Somatic and posttraumatic symptoms mediate the association between teachers’ victimization and burnout’. Could you please justify why did you include mediators in your study. Are there inconsistencies in the previous findings reported for your IV and DV. Why did you choose somatic and posttraumatic symptoms as mediators. Thus, kindly support with literatures in structural manner especially e.g IV dan mediator, mediator & DV.
3. Software used: IBM SPSS and SAS version 9.4 (CALIS procedure). Could you justify how this software compatible to analyse SEM.
4. Could you please recheck results part, the numbers that you reported (e.g Half of the participating teachers identified as Arabs (56.4%), less than half as Jews (43%), and a small pro-portion (6%) did not indicate their ethnicity- Total percentage more than 100?
5. As the 1st-2nd points above are my major concerns. The other rest of your writing is affected accordingly. Thus, I do not pin point them yet in specific until my concerns above are fulfilled.
Author Response
Dear Reviewer 1,
We are resubmitting a manuscript titled " Teachers’ Workplace Victimization, Job Burnout, and Somatic and Posttraumatic Symptoms: A Structural Equation Modeling Examination", which received a decision on correction and resubmission.
We carefully revised the manuscript based on your suggestions and included all necessary changes and adjustments. We found that this feedback greatly improved the manuscript.
Below are the changes we applied to align with your comments, point by point. We hope we have responded adequately.
Reviewer 1: You intend to establish SEM but I did not find any prominent theory in your article to support your conceptual/theoretical framework (Kindly refer Hair et al. & etc.)
Your 2nd hypothesis- ‘Somatic and posttraumatic symptoms mediate the association between teachers’ victimization and burnout’. Could you please justify why did you include mediators in your study. Are there inconsistencies in the previous findings reported for your IV and DV. Why did you choose somatic and posttraumatic symptoms as mediators. Thus, kindly support with literatures in structural manner especially e.g IV dan mediator, mediator & DV.
Authors: Thank for these important and critical comments. To address them, we further elaborated on the associations between the study variables and the conceptual foundations (pp. 2–3). As suggested, we referred to Hair to support the use of the SEM technique to better understand and explain the theoretical models (pp. 4–5).
These parts read as follows (pp. 2–3):
1.3. Associations of Teachers’ Workplace Victimization, Posttraumatic Symptoms, and BurnoutThe scientific literature linking TWV, posttraumatic and somatic symptoms, and burnout revealed inconclusive evidence. Prior studies indicated a direct association be-tween TWV and burnout [43–46]. Teachers experiencing workplace victimization were also more likely to be absent from school and leave the teaching profession—clear indications of job burnout [1,20,25,35,36].Other studies suggested that teachers’ and other professionals’ workplace victimization is associated with somatic symptoms, PTSD symptoms, and burnout [9,14–20,42]. Thus, we hypothesized that TWV will be directly associated with symptoms and burnout.Other studies suggested an indirect association between workplace victimization and burnout via the mediation of posttraumatic and somatic symptoms (e.g., Yang [47]). To illustrate, Berg and Cornell [5] identified an association between teacher-directed violence and burnout, with some burnout explained by distress symptoms. Similarly, Rojas-Flores and colleagues [31] indicated a significant indirect effect of teachers’ exposure to violence on burnout through PTSD symptomatology across two independent samples of teachers in Central America. Yet a different study indicated a nonsignificant association between burnout and symptoms, suggesting that teachers’ victimization via occasional and severe bullying predicted psychological distress, although burnout did not mediate those associations [48]. These inconclusive and contradictory findings demand additional investigation. Thus, we further tested the indirect association between TWV and burnout via the mediation of symptoms.The paragraph regarding the use of SEM appears in the literature review, just before the study hypothesis, and reads as follows (pp. 4–5):In summary, prior findings demonstrate inconclusive evidence of the associations among teachers’ background characteristics, conflictual work relationships, TWV, somatic and posttraumatic symptoms, and job burnout. Because the structure of relationships between the variables remain unclear this study sought to fill this gap by estimating and testing the structure of pathways among TWV and related outcomes as presented in the theoretical model that was developed based on the existing literature. Figure 1 presents the full study theoretical model.A theoretical model visually displays structural relationships, typically based on equations connecting conceptual variables, to formalize a theory. SEM involves statistical methods that estimate relationships among observed variables that represent conceptual variables in statistical models [84].
Reviewer 1: Software used: IBM SPSS and SAS version 9.4 (CALIS procedure). Could you justify how this software compatible to analyses SEM.
Authors: To address this important comment, we added a reference supporting the use of IBM SPSS software for conducting SEM investigations (Blunch, 2012).
Reviewer 1: Could you please recheck results part, the numbers that you reported (e.g Half of the participating teachers identified as Arabs (56.4%), less than half as Jews (43%), and a small pro-portion (6%) did not indicate their ethnicity- Total percentage more than 100?
Authors: Thank you for noticing this typo. We corrected the sentence, which now reads as follows (p. 8): “A small proportion (0.6%) did not indicate their ethnicity.”
Reviewer 1: As the 1st-2nd points above are my major concerns. The other rest of your writing is affected accordingly. Thus, I do not pin point them yet in specific until my concerns above are fulfilled.
Authors: We are happy to further respond to any concerns that will help improve our manuscript.
We wish to thank you again for your thorough review and helpful suggestions and comments, which helped improving the manuscript.
Appreciatively,
The authors

Reviewer 2 Report
Thank you for giving me the opportunity to review this paper. I found the work innovative and interesting. The analysis starts from a comprehensive study of the literature.
I would suggest further updating the literature by including citations of even more recent articles than those mentioned. For example, personal variables should also include self-esteem in the introduction (see for example Cataudella, S., Carta, S. M., Mascia, M. L., Masala, C., Petretto, D. R., Agus, M., & Penna, M. P. (2021). Teaching in times of the COVID-19 pandemic: A pilot study on teachers' self-esteem and self-efficacy in an Italian sample. International Journal of Environmental Research and Public Health, 18(15), 8211). I would also include in the conclusions the need to carry out possible cross-cultural studies and to involve teachers of all levels.
Author Response
Dear Reviewer 2,
We are resubmitting a manuscript titled " Teachers’ Workplace Victimization, Job Burnout, and Somatic and Posttraumatic Symptoms: A Structural Equation Modeling Examination", which received a decision on correction and resubmission.
We carefully revised the manuscript based on your suggestions and included all necessary changes and adjustments. We found that this feedback greatly improved the manuscript.
Below are the changes we applied to align with your comments, point by point. We hope we have responded adequately.
Reviewer 2: I would suggest further updating the literature by including citations of even more recent articles than those mentioned. For example, personal variables should also include self-esteem in the introduction (see for example Cataudella, S., Carta, S. M., Mascia, M. L., Masala, C., Petretto, D. R., Agus, M., & Penna, M. P., 2021). Teaching in times of the COVID-19 pandemic: A pilot study on teachers’ self-esteem and self-efficacy in an Italian sample. International Journal of Environmental Research and Public Health, 18(15), 8211). I would also include in the conclusions the need to carry out possible cross-cultural studies and to involve teachers of all levels.
Authors: Your suggestion is greatly appreciated, and we agree that self-esteem could contribute to teachers’ workplace victimization, symptoms, and burnout. Yet unfortunately, the study has already been carried out; thus, we do not have information on the teachers’ self-esteem and self-efficacy. However, because these are very important variables to examine, we added a recommendation for future studies to include these variables. We also recommended conducting cross-cultural studies to develop knowledge on how TWV, burnout, and symptoms may change depending on cultural affiliation and to include teachers working with students of all ages to allow a comparison across grade levels. This part reads as follows (p. 11):
We further recommend examining more deeply ethnocultural disparities in TWV, antecedents, and outcomes across Arabic and Jewish societies and different roles like educator, manager, and professional teacher to determine if differential relationships exist between roles in the two societies and burnout and symptoms. Additional research is encouraged to examine the associations among TWV, burnout, and symptoms across different cultures to uncover whether their associations manifest differently depending on the cultural context. Because school violence manifests differently across grade levels, additional research could explore differences in the study model among teachers working in schools of all grade levels.
The current findings are unique in that they suggest that teachers who experience victimization and develop symptoms experience burnout more than teachers who do not develop symptoms. Nonetheless, because the data were collected during the outbreak of COVID-19 in Israel, educators likely experienced increased stress that affected the measurement. Thus, we encourage researchers to study TWV, job burnout, and symptoms in regular and less stressful periods. Further, because research has pointed to the critical role of teachers’ self-efficacy and self-esteem in times of crisis [108], we recommend including these variables in future studies.
We wish to thank you again for your thorough review and helpful suggestions and comments, which helped improving the manuscript.
Appreciatively,
The authors
